# Estimation of an Exposure Threshold Value for Compensation of Silica-Induced COPD Based on Longitudinal Changes in Pulmonary Function

**DOI:** 10.3390/ijerph17239040

**Published:** 2020-12-04

**Authors:** Matthias Möhner, Dennis Nowak

**Affiliations:** 1Division Work and Health, Federal Institute for Occupational Safety and Health, D-10317 Berlin, Germany; 2Institute and Clinic for Occupational, Social and Environmental Medicine, Clinical Centre of the Ludwig Maximilian University Munich, D-80539 Munich, Germany; Dennis.Nowak@med-uni-muenchen.de; 3Comprehensive Pneumology Center (CPC) Munich, German Center for Lung Research (DZL), D-81377 Munich, Germany

**Keywords:** chronic obstructive pulmonary disease, respirable crystalline silica, threshold models, epidemiology, compensation

## Abstract

(1) Background: To estimate the cumulative exposure to respirable crystalline silica (RCS) that reduces lung function to an extent corresponding with airway obstruction equivalent to chronic obstructive pulmonary disease (COPD). (2) Methods: The study is based on a miners’ cohort with longitudinal data on lung function and RCS exposure. Random-effects linear regression models, allowing for a possible threshold concerning the exposure concentration were used to analyze the impact of RCS on the ratio of forced expiratory volume in 1 s and forced vital capacity (FEV_1_/FVC). The proposed threshold is the amount of RCS resulting in a decrease in FEV_1_/FVC from the expected value to the lower limit of normal. (3) Results: The analysis shows that a threshold model fits the data significantly better than the usual linear model. The estimated threshold value for the exposure concentration is 0.089 mg/m^3^. Using this threshold, the estimate for the corresponding reference dose for RCS is 2.33 mg/m^3^·y. (4) Conclusions: The analysis confirmed that RCS has a negative impact on lung function. The effect is primarily due to exposure above a concentration threshold of 0.1 mg/m^3^. It is recommended that COPD should be compensated as an occupational disease if cumulative exposure was at least 2 mg/m^3^·y above this threshold.

## 1. Introduction

Chronic obstructive pulmonary disease (COPD) is a major cause of chronic morbidity and mortality worldwide, especially in subjects over 40 years of age. Although tobacco smoking is the most important single risk factor for COPD, studies have shown that occupation contributes to 15 to 20% of COPD cases [1,2,3,4,5,6]. Respirable crystalline silica (RCS) is one of the most widespread occupational exposures associated with COPD. A meta-analysis has shown that occupational exposure to RCS was associated with a statistically significant decrease in forced expiratory volume in 1 s (FEV_1_) and FEV_1_/forced vital capacity (FVC), revealing airway obstruction consistent with COPD [7]. However, most of the studies included are cross-sectional and the results of the studies are usually not suitable for deriving quantitative estimates of the relationship between RCS exposure and COPD. One longitudinal study attempted to quantify the risk of developing COPD depending on cumulative RCS exposure, adjusting for smoking [8]. However, the outcome classification was based only on the pre-bronchodilator FEV_1_/FVC ratio, which led to an overestimation of the number of COPD cases, and thus to biased risk estimates. Consequently, these results cannot be used to derive a criterion for compensation in occupational diseases law. The aim of the present analysis is to develop a new methodological approach which can be used to derive such a criterion, and thus also to enable the recognition of RCS-related COPD as an occupational disease in Germany outside the coal mining industry.

## 2. Materials and Methods

We report this study according to the Strengthening the Reporting of Observational Studies in Epidemiology guidelines [9].

### 2.1. Study Population

The study population is a sub-cohort of the German uranium miners cohort study [10,11] and was already used in a previous analysis [8]. In brief, the sub-cohort consists of subjects born between 1954 and 1956 to ensure that none of the study subjects had worked underground in the mines before 1971. The reason for this restriction was that by 1970, the working conditions had improved to a point where international standards of industrial hygiene and radiation protection were being met. Further details of the cohort composition are described elsewhere [8,10].

Ethics Approval: The data protection officer of the Federal Institute for Occupational Safety and Health and the Institutional Review Board approved the study design.

### 2.2. Lung Function Data

The lung function data were taken from the documentation of the occupational health check-ups carried out every 2 years [12]. Particularly because of the high incidence of silicosis among those miners who worked in uranium ore mining in the early post-war years, the occupational health examination focused on the early detection of respiratory diseases. For this purpose, a special spirometer was even developed by the mining company (Stollberg spirometer) and was used throughout the entire study period. The occupational medical examination was usually carried out before the morning shift. The pulmonary function testing (PFT) was performed without the administration of a bronchodilator.

### 2.3. Exposure Assessment

Detailed information about miner’s occupational histories comprises data on mining facility/shaft, job type, number of shifts, and periods of absence on a daily basis. The German Statutory Accident Insurance (DGUV) collected these data from payrolls and personal files [10]. In combination with a comprehensive job-exposure matrix, based on extensive site-by-site measurements for the original mine equipment until the 1980s [13], estimates of the cumulative RCS exposure for each miner until the dates of each medical check-up performed were calculated. Further details are described elsewhere [8].

### 2.4. Statistical Methods

The major criterion for the definition of COPD is based on post-bronchodilator FEV_1_/FVC. According to the GOLD criteria, a COPD is assumed if FEV_1_/FVC is below 0.7 [14]. However, normal spirometry may be defined by an approach from the Global Lung Initiative (GLI), taking into account ethnicity, gender, body height, and spirometric variability across the lifespan [15]. Based on more than 70,000 records of healthy nonsmokers, a suitable power transformation for the distribution of the ratio FEV_1_/FVC was derived using the LMS method [15,16,17]. The distribution of the transformed value, defined by
(1)$$z=\frac{{\left(\frac{FEV_{1}}{FVC}/\mu \right)}^{\lambda }-1}{\lambda \cdot \sigma }$$
and referred to as *z*-score, is close to the standard normal distribution. Here, λ represents the skewness, μ the expected value and σ the coefficient of variation in the distribution function. The GLI diagnostic approach has a cut-off at the lower level of normal (LLN) values for the ratio FEV_1_/FVC, where LLN classifies the bottom 5% of the healthy population as abnormal. Both approaches have their advantages and disadvantages. For younger subjects, LLN is above 0.7, but below 0.7 for people over 50 years of age. Considering that for our cohort, only pre-bronchodilator data are available, we prefer the GLI approach and we use the corresponding *z*-score for the subsequent analysis.

In our methodological approach, we assume that a COPD can be regarded as RCS-induced if RCS has caused a decrease in FEV_1_/FVC from the expected value μ down to the level LLN or, equivalently, the *z*-score, to decrease by 1.645 (from 0 to −1.645). Thus, a reference dose for RCS can be estimated knowing the average decrease in *z* score per unit RCS. Because the *z*-score already takes into account the impact of age, gender, ethnicity, and body height, it seems plausible to consider *z* only as a function of RCS exposure and smoking status. Random-effects linear regression models were used to analyze the impact of RCS exposure on *z*. The class of models was further extended to allow for a possible threshold for the concentration of RCS using the approach described by Ulm [18]. Hence, ***z*** can be modelled by
(2)$$z_{it}=\beta_{0}-\beta_{1}\times RCS_{it}+\beta_{2}\times SMOKER_{i}+u_{0i}+\varepsilon_{ij}$$
where i=1, …,n denotes the cohort subject and $t=1,\;\ldots ,\;i_{t}$ the number of PFT. $RCS_{it}\left(\tau \right)$ denotes the cumulative exposure in the relevant time period above the threshold τ, i.e., $RCS_{it}\left(\tau \right)=\sum_{j=1}^{t}\left[\left(time_{i,j}-time_{i,j-1}\right)\times I_{\left(c_{ij}\geq \tau \right)}\times \left(c_{ij}-\tau \right)\right]$, and $I_{\left(c_{ij}\geq \tau \right)}$ is the corresponding indicator function regarding the average RCS concentration $c_{ij}$ for subject i in the respective time period, with the values 0 and 1. $SMOKER_{i}$ are binary variables, which indicate subject’s smoking status (never/ever). The term $u_{0i}$ was introduced as a random intercept for miner i.

All (integer) concentration values between 0 and 160 µg/m^3^ were assumed as potential threshold values. By including the “0”, the model without threshold value is also included in the comparison. A 95% confidence interval for the estimator $\hat{\tau }$ can be determined using the likelihood ratio test statistics $D\left(\tau \right)=2\times \left(\ln L\left(\hat{\tau }\right)-\ln L\left(\tau \right)\right)<\chi_{1,0.95}^{2}$, which follows a χ^2^-distribution with one degree of freedom [18]. The reference dose, hereinafter referred to as RCS_LLN_(τ), is then obtained as the quotient of 1.645 and $\beta_{1}$. It represents an estimate of the cumulative RCS dose, which, on average, results in a reduction in FEV_1_/FVC from the predicted value to LLN.

In the previous analysis [8], instead of the pure smoking status, the years of smoking were taken into account. To allow a better comparison with these results, the following model was used for comparison
(3)$$z_{it}=\beta_{0}-\beta_{1}\times RCS_{it}+u_{0i}+u_{1i}\times syr_{it}+u_{2i}\times usyr_{it}+\varepsilon_{it}$$

The number of years a miner smoked between the first and t-th spirometry for a continuous smoker and a miner with uncertain smoking status are denoted by $syr_{it}$ and $usyr_{it}$, respectively. Moreover, random slopes $u_{1i}$ and $u_{2i}$ were considered to account for differences between miners regarding intensity of tobacco consumption. Model parameters were estimated by mixed-effects maximum-likelihood regression.

Akaike’s information criterion as well as the Bayesian information criterion were used to compare goodness of fit for models [19,20]. All statistical analyses were performed using the STATA package release 16.1 [21]. A uniform significance level of α = 0.05 was used for statistical tests.

## 3. Results

Three subjects had to be excluded from the original database because of a wrong match between individual exposure and health data. Therefore, the final database contains 7116 data records for 1418 miners in the age range from 18 to 36 years. More than half of the miners were younger than 20 years at hire. Due to the closure of the East German uranium mines by the end of 1990, the age at last PFT by the occupational health check-ups was not more than 36 years. For 9.3% of the cohort, the first recorded value for FEV_1_/FVC was below LLN, for 5.1% even below 0.7. None of the miners were diagnosed with silicosis. Further characteristics of the cohort are given elsewhere [8].

Analysis of the complete dataset shows that a threshold model fits the data significantly better than the usual linear model. The threshold value for the exposure concentration is estimated to be 0.089 mg/m^3^ (95%CI: 0.071 to 0.101 mg/m^3^). Applying this threshold leads to a significant improvement in model fitting compared to the model without threshold value. In this model the *z*-score is reduced by 0.706 (95%CI: 0.528 to 0.884) per 1 mg/m^3^·y of cumulative RCS exposure above this threshold. Accordingly, the estimate for the reference dose $RCS_{LLN}\left(0.089\right)$ is 2.33 (95%CI: 1.86 to 3.11) mg/m^3^·y. The corresponding results for some sub-cohorts are given in Table 1. Since the estimates for the threshold value are different in the sub-cohorts, the estimates for the reference dose were additionally calculated based on a uniform threshold value τ = 0.1 mg/m^3^ in order to enable better comparability of the results. Some exposure characteristics of the investigated sub-cohorts are shown in Table 2. The highest gradient in terms of RCS exposure was determined for miners who had stopped mining uranium before 9 November 1989 (with the fall of the Berlin Wall on 9 November 1989 and the associated political changes, it quickly became clear that uranium ore mining would soon come to an end).

The consideration of time of smoking led to a significantly better goodness of fit for the model [3]. However, the change in risk estimates concerning RCS is marginal in comparison to model [2] (see Table 3). However, due to the higher number of model parameters, the respective confidence intervals are wider.

## 4. Discussion

Based on an inception cohort comprising more than 1400 miners with a follow-up of, on average, more than 12 years and five pulmonary function tests, evidence was found for an exposure–response relationship between cumulative RCS exposure and pre-bronchodilator lung function with a threshold for a exposure concentration of about 0.1 mg/m^3^. The analysis of the cohort, restricted to those miners with an average exposure concentration of less than 0.1 mg/m^3^ in all time periods between two consecutive PFTs (N = 797), supports the existence of a threshold. The estimate for the corresponding parameter $\beta_{1}$ is only 0.060 (95%CI: −0.066 to 0.186) and it is not significantly elevated (*p* > 0.3). Finally, if we only consider those miners (N = 386) whose exposure concentration is below the current occupational exposure limit (OEL) of 0.05 mg/m^3^ in Germany, the estimate for $\beta_{1}$ drops to 0.051 (95%CI: −0.318 to 0.420; *p* > 0.7). Thus, our study supports the conclusion drawn from a review of longitudinal studies in a number of industries that loss of lung function occurs with exposure to silica dust at concentrations of 0.1 mg/m^3^ and above [22]. It should also be noted that a threshold of 0.1 mg/m^3^ for compensation claims is not in conflict with the OEL. The latter is below almost all of the confidence intervals given in Table 1, which we consider necessary for the purpose of prevention.

### 4.1. Strengths and Weaknesses

A major strength of our study is that reliable exposure data are available for all study subjects. Moreover, the pulmonary function tests took place under largely comparable framework conditions with the same equipment and with centrally trained staff throughout the study period.

We are aware of some weaknesses of our study. Our lung function data do not extend beyond the age of 36, i.e., they are well ahead of the time at which COPD clinically manifests, on average. This can lead to a bias in the extrapolation of the results into older age groups. The early observation phase may also have led to an underestimation of the impact of smoking. The analysis of the non-smokers showed that, for them, a significantly weaker decline in lung function parameters was observed. On the other hand, the estimated reference dose increases with cumulative exposure (Table 1 and Table 3). In addition, a higher reference dose was also calculated for miners who worked until the mines were closed, and thus worked much longer than those who left early. Another important issue is the definition of COPD in our study. In the previous analysis, a prevalence of COPD of 16.4% (stage I+, GOLD) and 9.3% (stage II+, GOLD) had been reported based solely on pre-bronchodilator lung function values. On the basis of the initial PFTs (mean age 23 years), the prevalence would have been 5.1% (stage I+, GOLD) and 3.0% (stage II+, GOLD) or even 9.3% (stage I+, GLI).

Data on the prevalence of COPD in the age group below 40 are scarce. From the Third National Health and Nutrition Examination Survey a prevalence for COPD (stage II+, GOLD) of 1.7% was reported in the 30–34 age group [23]. Furthermore, in a European cohort study, the incidence of COPD (stage I+, GOLD) in the age group 20–30 years was estimated at 1.5 per 1000 personyears [24]. The considerable effects of using the pre- instead of the post-bronchodilator values were also evident from other studies [25,26].

The use of a COPD definition based solely on pre-bronchodilator spirometry has thereby led to an overestimation of the prevalence in the previous analysis. To mitigate this issue, we use, in our present approach, only the estimates for the RCS-induced declines in lung function instead of the lung function parameters itself. We assume that this decline can also serve as a good approximation for the decline in post-bronchodilator lung function due to RCS, and that it is independent of the clinical symptoms of COPD (such as cough, sputum and shortness of breath).

### 4.2. Healthy Worker Effect

From the results presented in Table 2, the significantly stronger decrease for miners who left uranium ore mining early is striking, which, in turn, suggests an HWE. However, the comparison of these miners with those who were still employed after the cut-off date shows that the former already had significantly worse lung function at the first spirometry, corresponding to 4.3 percentiles. This difference increased to 4.9 percentiles by the last available spirometry, even though the cumulative RCS exposure of the miners who left the mine early under virtually identical exposure conditions and the exposure time was one third lower than for the other miners (Table 4). The significantly higher proportion of smokers in this group suggests that tobacco consumption may be the main cause of this apparent HWE, especially since bonus cigarettes were an integral part of the wage system in uranium mining. Other studies have also shown that heavy smokers already have a four-fold risk of COPD in young adults compared to non-smokers [24].

Unfortunately, quantitative data on smoking were not available for the cohort. In this respect, the differences between the estimates for regular smokers and non-smokers are likely to underestimate the real effect of smoking.

### 4.3. Alternative Model Approaches

In the approach described so far, the ***z*** score is used as the underlying metric. Similarly, the ratio FEV_1_/FVC can be used without transformation. Analogous to models [2] and [3], we can now consider a model with $∆=(FEV_{1}/FVC)-\mu$ instead of ***z*** as the dependent variable. The coefficient for the decrease in ∆ is then set in relation to the difference between the predicted value μ and LLN, i.e., about 0.104 in the age range from 20 to 70 years. The corresponding results differ only slightly from those of the approach described above. For example, under model [2], taking into account the entire data pool *τ* = 0.090 mg/m^3^ is estimated as the optimal threshold value and from *τ* = 0.1 results RCS_LLN_ (0.1) = 1.96 mg/m^3^∙y (95%CI: 1.54–2.69).

Another possibility would be to convert the *z* score into the corresponding percentile using the distribution function of the standard normal distribution, i.e., Φ(z)×100 in the value range (0, 100) is examined. In this metric, the difference between the predicted value and LLN is 45 (the predicted value corresponds to 50 and LLN to 5). However, when comparing the goodness of fit, this approach scores worse than the other two approaches.

### 4.4. Practical Examples

A large study showed that COPD was significantly more common in miners (33%) than non-miners (26%) [27]. However, no quantitative data on RCS exposure were available. It is therefore difficult to make a decision under these conditions as to which of the COPD cases among the miners is to be recognized as an occupational disease. The exposure concentration can be very different, even if the workers perform similar work. For example, the range of exposure concentration in 90 Norwegian tunnel workers varied between 0.005 and 1.041 mg/m^3^ [28].

The above analyses show that the negative impact of RCS is primarily due to exposure above a concentration threshold of about 0.1 mg/m^3^. An average of 2 mg/m^3^·y RCS above this concentration threshold leads to a decrease in the ratio FEV_1_/FVC to the level of LLN. In a study on Swedish granite crushers, the mean duration of exposure was 22 years and the mean cumulative exposure was 7 mg/m^3^ [29]. Hence, the mean concentration was 0.32 mg/m^3^, i.e., 0.22 mg/m^3^ above the estimated concentration threshold. Thus, assuming a constant concentration of 0.32 mg/m^3^, a granite crusher would have reached the reference dose of 2 mg/m^3^·y RCS after 9 years and one month. If he was diagnosed with COPD at this time or later, it would have to be recognized as occupational. A recent study among Norwegian rock drillers serve as another example [30]. Among the listed job categories, only the rock driller using feed mounted control panel (0.24 mg/m^3^ RCS) was exposed above the threshold level. He would reach the reference dose after 16 years and 8 months of constant exposure at this level, so that a potential COPD, in his case, would be considered occupational from that time on.

## 5. Conclusions

The present analysis has confirmed that RCS has a negative impact on lung function. However, the effect is primarily due to exposure above a concentration threshold of about 0.1 mg/m^3^. An average of 2 mg/m^3^·y RCS above this concentration threshold leads to a decrease in the ratio FEV_1_/FVC to the level of LLN, although this cumulative measure should be regarded as a conservative estimate. It is therefore recommended to recognize a COPD according to the GOLD definition [14] as an occupational disease if the cumulative RCS-exposure above the concentration threshold of 0.1 mg/m^3^ is more than 2 mg/m^3^·y.

## Figures and Tables

**Table 1 ijerph-17-09040-t001:** Estimates for the reference dose of respirable crystalline silica based on threshold models and random effects maximum-likelihood regression (95%CI in parentheses).

Subcohort	n_min_	N	τ	Based on Best Estimate for τ		Based on τ = 0.1	
				β	${\mathrm{RCS}}_{\mathrm{LLN}}(\tau )$	β	RCS_LLN_(0.1)
	2	1418	0.000	0.165	9.96 (7.51 to 14.8)		
	2	1418	0.089 (0.071 to 0.101)	0.706	2.33 (1.86 to 3.11)	0.840	1.96 (1.56 to 2.64)
	5	721	0.092 (0.077 to 0.105)	0.755	2.18 (1.70 to 3.02)	0.862	1.91 (1.49 to 2.66)
	8	268	0.094 (0.073 to 0.116)	0.731	2.25 (1.58 to 3.90)	0.820	2.01 (1.41 to 3.49)
Non-smokers	2	231	0.076 (0.000 to 0.116)	0.534	3.08 (1.99 to 6.79)	0.664	2.48 (1.56 to 6.04)
Regular smokers	2	867	0.130 (0.071 to 0.147)	1.522	1.08 (0.81 to 1.64)	0.891	1.85 (1.37 to 2.82)
≥ 0.5 mg/m^3^∙y	2	824	0.094 (0.077 to 0.109)	0.680	2.42 (1.87 to 3.43)	0.746	2.20 (1.70 to 3.14)
≥ 1.0 mg/m^3^∙y	2	441	0.092 (0.068 to 0.111)	0.576	2.86 (2.10 to 4.46)	0.645	2.55 (1.87 to 4.03)
≥ 1.5 mg/m^3^∙y	2	184	0.093 (0.047 to 0.116)	0.600	2.74 (1.95 to 4.63)	0.660	2.49 (1.77 to 4.22)
Early exit	2	450	0.072 (0.039 to 0.094)	0.759	2.17 (1.54 to 3.66)	1.135	1.45 (0.99 to 2.71)
Later exit	2	968	0.091 (0.074 to 0.105)	0.701	2.35 (1.82 to 3.31)	0.807	2.04 (1.57 to 2.90)
max(c) < 0.1 mg/m^3^	2	797	0.000	0.060	27.5 (8.9 to ∞)		

n_min_—Required minimum number of records per subject. N—Number of subjects included in the model. τ—Threshold value for exposure concentration of respirable crystalline silica (mg/m^3^). **β**—Coefficient for the reduction in the *z*-score of. FEV_1_/FVC per 1 mg/m^3^∙y above τ. RCS_LLN_ (τ)—Reference dose for the reduction of the FEV_1_/FVC down to LLN for a given threshold value τ. Early exit—Termination of work in the uranium mines no later than 9 November 1989. max(c)—Maximum exposure concentration.

**Table 2 ijerph-17-09040-t002:** Some characteristics of subgroups (means (interquartile ranges) or percentages).

Sub-Cohort	N	Duration of Employment (Years)	Mean Concentration of RCS (mg/m^3^)	Mean Cumulative Exposure (mg/m^3^∙y)	Smoking Status (%)		
					Never	Regular	Intermittent
All miners	1418	12.8 (9.3 to 16.8)	0.072 (0.034 to 0.099)	0.764 (0.288 to 1.140)	16.3	61.1	22.6
Non-smokers	231	13.2 (9.9 to 17.3)	0.072 (0.025 to 0.100)	0.745 (0.275 to 1.052)	100	0	0
Regular smokers	867	12.4 (8.1 to 16.5)	0.073 (0.038 to 0.101)	0.747 (0.270 to 1.125)	0	100	0
n_min_ ≥ 5	721	14.9 (13.3 to 17.3)	0.075 (0.044 to 0.098)	0.971 (0.531 to 1.346)	15.0	57.7	27.3
n_min_ ≥ 8	268	16.3 (14.6 to 18.3)	0.074 (0.051 to 0.098)	1.112 (0.727 to 1.504)	11.6	53.3	35.1
≥0.5 mg/m^3^∙y	824	14.6 (12.8 to 17.3)	0.093 (0.070 to 0.109)	1.129 (0.750 to 1.464)	15.3	60.3	24.4
≥1.0 mg/m^3^∙y	441	16.2 (14.7 to 18.3)	0.105 (0.087 to 0.115)	1.465 (1.177 to 1.683)	14.5	59.2	26.3
≥1.5 mg/m^3^∙y	184	17.0 (15.9 to 18.3)	0.120 (0.101 to 0.132)	1.799 (1.600 to 1.857)	18.5	56.0	25.5
Early exit	450	9.0 (5.3 to 12.6)	0.073 (0.033 to 0.102)	0.536 (0.193 to 0.772)	13.6	68.9	17.5
Later exit	968	14.6 (13.0 to 17.3)	0.072 (0.034 to 0.098)	0.870 (0.361 to 1.281)	17.6	57.5	24.9
Max(c) < 0.1 mg/m^3^	797	12.2 (8.0 to 16.2)	0.046 (0.022 to 0.065)	0.445 (0.203 to 0.628)	17.2	59.7	23.1

N—Number of subjects included in the subgroup. Early exit—Termination of work in the uranium mines no later than 9 November 1989. max(c)—Maximum exposure concentration over all exposure periods.

**Table 3 ijerph-17-09040-t003:** Estimates for the reference dose of respirable crystalline silica based on threshold models using mixed-effects maximum-likelihood regression (95%CI in parentheses).

Subcohort	n_min_	N	τ	Based on Best Estimate for τ		Based on τ = 0.1	
				β	${\mathrm{RCS}}_{\mathrm{LLN}}(\tau )$	β	RCS_LLN_(0.1)
	2	1418	0.000	0.180	9.12 (6.22 to 17.06)		
	2	1418	0.084 (0.065 to 0.106)	0.573	2.87 (2.16 to 4.28)	0.701	2.35 (1.75 to 3.56)
	5	721	0.084 (0.047 to 0.102)	0.642	2.56 (1.91 to 3.88)	0.769	2.14 (1.58 to 3.31)
	8	268	0.084 (0.017 to 0.116)	0.581	2.83 (1.83 to 6.31)	0.703	2.34 (1.49 to 5.46)
Non-smokers	2	231	0.077 (0.007 to 0.121)	0.543	3.03 (1.94 to 6.90)	0.670	2.46 (1.54 to 6.15)
Regular smokers	2	867	0.131 (0.061 to 0.152)	1.245	1.32 (0.90 to 2.51)	0.731	2.25 (1.49 to 4.56)
≥ 0.5 mg/m^3^∙y	2	824	0.094 (0.073 to 0.133)	0.553	2.97 (2.11 to 5.02)	0.594	2.77 (1.96 to 4.70)
≥ 1.0 mg/m^3^∙y	2	441	0.093 (0.055 to 0.144)	0.447	3.68 (2.37 to 8.20)	0.476	3.46 (2.22 to 7.83)
≥ 1.5 mg/m^3^∙y	2	184	0.082 (0.000 to 0.137)	0.397	4.15 (2.46 to 13.2)	0.448	3.68 (2.16 to 12.4)
Early exit	2	450	0.072 (0.023 to 0.133)	0.644	2.55 (1.59 to 6.41)	0.866	1.90 (1.13 to 6.03)
Later exit	2	968	0.089 (0.064 to 0.109)	0.604	2.72 (1.99 to 4.33)	0.687	2.39 (1.73 to 3.86)
max(c) < 0.1 mg/m^3^	2	797	0.000	0.052	31.5 (7.18 to ∞)		

n_min_—Required minimum number of records per subject. N—Number of subjects included in the model. τ—Threshold value for exposure concentration of respirable crystalline silica (mg/m^3^). **β**—Coefficient for the reduction of the ***z*** score of FEV_1_/FVC per 1 mg/m^3^∙y above τ. RCS_LLN_ (τ)—Reference dose for the reduction in the FEV_1_/FVC down to LLN for a given threshold value τ. Early exit—Termination of work in the uranium mines not later than 9 November 1989. max(c)—Maximum exposure concentration

**Table 4 ijerph-17-09040-t004:** Mean and 95%CI for FEV_1_/FVC and related parameters for early vs. late exit from mining job.

Spirometry	Parameter	Early Exit	Late Exit
	N	450	968
	Never-smokers (%)	13.6	17.6
	Regular smokers (%)	68.9	57.5
First	RCS (mg/m^3^∙y)	0.13 (0.11 to 0.15)	0.22 (0.20 to 0.24)
	FEV_1_/FVC-µ (%)	−2.42 (−1.79 to −3.07)	−1.65 (−1.19 to −2.11)
	*z* score	−0.31 (−0.21 to −0.40)	−0.18 (−0.11 to −0.25)
	P(z)	41.6 (39.0 to 44.2)	45.9 (44.0 to 47.8)
Last	Elapsed time (y)	5.6 (5.2 to 5.9)	8.8 (8.5 to 9.1)
	RCS (mg/m^3^∙y)	0.54 (0.49 to 0.58)	0.87 (0.83 to 0.91)
	FEV_1_/FVC-µ (%)	−3.53 (−2.85 to −4.21)	−2.41 (−1.91 to −2.90)
	*z* score	−0.48 (−0.38 to −0.58)	−0.31 (−0.23 to −0.38)
	P(z)	37.7 (35.0 to 40.3)	42.6 (40.7 to 44.6)

µ—Expected value of FEV_1_/FVC, depending on age and body height (%). P(z)—Percentile of z.

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
