# Peer review of "Estimation of an Exposure Threshold Value for Compensation of Silica-Induced COPD Based on Longitudinal Changes in Pulmonary Function"

_ijerph, 2020, doi:10.3390/ijerph17239040_

Round 1

Reviewer 1 Report

Overall this is study is well designed with data from 1418 miners who had RCS exposure and PFT's performed. In the statistical method sections, authors mentioned that they prefer the GLI approach to diagnose COPD and use the corresponding z score for the 90 subsequent analysis. They comment on the concern of overestimation due to only have pre-bronchodilator PFT's and use GOLD criteria for PFT diagnosis.

However in the final discussion, they recommend to use the GOLD 252 definition as an occupational disease, and hope they can clarify about this conflicting comments. 

There is lack of quantitative data on the cigarette smoking which could have influenced the results. 

Author Response

For the miners only the pre-bronchodilator PFT’s were available, so that we could not check the GOLD-criteria completely. But usually only pre-bronchodilator PFT’s are available for routine occupational health examinations.

We assume that the decrease of lung function by RCS is nearly the same on the basis of pre-bronchodilator PFT’s and post-bronchodilator PFT’s.

Yes, there is lack of quantitative data on cigarette smoking. However, we have also applied mixed models, including random slopes for the smoking variable (see Table 3). Hence, it can be assumed that the slope is larger for a heavy smoker.

Reviewer 2 Report

The analysis is original and well elaborated. As already indicated the analysis is on young workers and no diseases was apparent yet at the time of the last spirometry. It would have been a tremendous but valuable effort to track down and check COPD status of a sub-group to confirm your predictions.

I find it a bit misleading to use the o.1 mg/m3 simply based on the value of the best thresholded fit, although it is close to current OEL of RCS, which is set to prevent silicosis and lung cancer. May be few words on a common AOP for all endpoints can be added to the discussion.

The outcomes are not new, but the cohort is rather unique and therefore I suggest to publish the paper, provided that there is some rewording of the 0.1 mg/m3 threshold and a statement in the abstract that no actual COPD was observed but that this is a prediction analysis. Therefore the title also needs amendment...

Author Response

We would very much like to follow-up the miners further, but for reasons of data protection we are not allowed to do so. We can only work with anonymous data.

The estimated reference dose increases with cumulative exposure (Tables 1 and 3). In addition, a higher reference dose was also calculated for miners who worked until the mines were closed and thus worked much longer than those who left early. Therefore, we assume that our estimate is conservative enough, even assuming longer exposure times. For example, the subgroup with a cumulative exposure ≥ 1.5 mg/m³âˆ™yrs had an average length of employment of 17 years. This is more than in many other studies.

Our estimated threshold value is only designed for the recognition of a COPD as an occupational disease. It has no direct relation to an OEL. However, an OEL must always be below a threshold value for the recognition of an occupational disease, because the OEL is supposed to protect against the disease.  We have included examples of the application of our method as Section 4.4 and hope that this will make it easier to understand.

Reviewer 3 Report

In my modest opinion this paper is not useful in daily practice since it is not applicable in the process of recognition and compensation of COPD, given that tha majority of work-related COPD cases are found in old former-workers that were exposed to RCS many years ago in different working sectors.  Therefore, it is necessary to perform a retrospective recostruction of working exposures based on estimations from available epidemiological data, that is without data about the exact concentration (mg/m3).  

In conclusion,  although the present study has some merits (i.e. scientific soundness and fair quality of presentation), as an occupational physician dealing with occupational diseases recognition, I do think that this paper is not sufficiently original and, overall, it is not of adequate interest to the readers.

Author Response

As seen in reference [27], the difference between miners and non-miners is not too big in terms of prevalence of COPD. Moreover, reference [28] shows that there are also large differences within an occupational group with regard to exposure concentration.

We believe that the knowledge of the exposure profile can help to make the fairest possible decision regarding the recognition of COPD as an occupational disease. We have included examples of the application of our method as Section 4.4 and hope that this will make it easier to understand.

Round 2

Reviewer 3 Report

The Authors have perfectly got the point of my previous review and I do think that the manuscript has been significantly improved accordingly.